# The role of manganese in CoMnOₓ catalysts for selective long-chain hydrocarbon production via Fischer-Tropsch synthesis

Hao Chen [1], Zan Lian[2], Xiao Zhao [3,4], Jiawei Wan [3,4], Priscilla F. Pieters [5], Judit Oliver-Meseguer[1], Ji Yang [3], Elzbieta Pach [3], Sophie Carenco [3], Laureline Treps[3], Nikos Liakakos[3], Yu Shan[1,4], Virginia Altoe [6], Ed Wong[6], Zengqing Zhuo [7], Feipeng Yang [7], Ji Su [3], Jinghua Guo [7], Monika Blum [7], Saul H. Lapidus[8], Adrian Hunt [9], Iradwikanari Waluyo [9], Hirohito Ogasawara [10], Haimei Zheng [3,4], Peidong Yang [1,4,5], Alexis T. Bell [1,11], Núria López [2] & Miquel Salmeron [1,3,4] ✉

Cobalt is an efficient catalyst for Fischer–Tropsch synthesis (FTS) of hydrocarbons from syngas (CO + H₂) with enhanced selectivity for long-chain hydrocarbons when promoted by Manganese. However, the molecular scale origin of the enhancement remains unclear. Here we present an experimental and theoretical study using model catalysts consisting of crystalline CoMnOₓ nanoparticles and thin films, where Co and Mn are mixed at the sub-nm scale. Employing TEM and in-situ X-ray spectroscopies (XRD, APXPS, and XAS), we determine the catalyst's atomic structure, chemical state, reactive species, and their evolution under FTS conditions. We show the concentration of CHₓ, the key intermediates, increases rapidly on CoMnOₓ, while no increase occurs without Mn. DFT simulations reveal that basic O sites in CoMnOₓ bind hydrogen atoms resulting from H₂ dissociation on Co⁰ sites, making them less available to react with CHₓ intermediates, thus hindering chain termination reactions, which promotes the formation of long-chain hydrocarbons.

The Fischer–Tropsch synthesis (FTS) reaction converts syngas (CO + H₂) to hydrocarbons, with cobalt (Co) being one of the most efficient catalysts[1–3], with the most desirable reaction products being long-chain hydrocarbons, rather than methane[4]. Numerous studies have shown that the product selectivity for C₅₊ and longer hydrocarbons can be enhanced by addition of manganese (Mn)[5–25]. Investigation by Weckhuysen and co-workers revealed the critical role played by MnO/Mn₁₋ₓCoₓO phases and by the reducible TiO₂ support

in improving selectivity towards long-chain hydrocarbons[7–10]. Bell et al. further elucidated the significance of the Lewis acid–base interaction in promoting the activation of CO at the Co-MnO interface[12–16,22]. Kruse et al. reported that formation of alcohols and aldehydes can be favored on CoMnOₓ catalysts[17,19–21], proposing a synergistic effect between Mn₅O₈ oxide and cobalt carbide (Co₂C) driving the reaction[17,21]. Sun et al. proposed that Co₂C is an active phase of CoMn catalyst for the selective conversion of syngas to

[1]Chemical Sciences Division, Lawrence Berkeley National Laboratory, Berkeley, CA, USA. [2]Institute of Chemical Research of Catalonia (ICIQ-CERCA), Barcelona Institute of Science and Technology (BIST), Tarragona, Spain. [3]Materials Science Division, Lawrence Berkeley National Laboratory, Berkeley, CA, USA. [4]Department of Materials Science and Engineering, University of California, Berkeley, CA, USA. [5]Department of Chemistry, University of California, Berkeley, CA, USA. [6]Molecular Foundry, Lawrence Berkeley National Laboratory, Berkeley, CA, USA. [7]Advanced Light Source, Lawrence Berkeley National Laboratory, Berkeley, CA, USA. [8]Advanced Photon Source, Argonne National Laboratory, Lemont, IL, USA. [9]National Synchrotron Light Source II, Brookhaven National Laboratory, Upton, NY, USA. [10]SLAC National Accelerator Laboratory, Menlo Park, CA, USA. [11]Department of Chemical and Biomolecular Engineering, University of California, Berkeley, CA, USA. ✉e-mail: mbsalmeron@lbl.gov

lower olefins, with Mn acting as a structure promoter in the formation of $Co_2C$ nano-prisms from the $Co_xMn_{1-x}O$ precursor[18,24]. So far however, the nature of the catalytic active phase in Co/Mn compounds, and the molecular scale mechanism of the FTS reaction is still under debate. To advance in this field, a better knowledge of the atomic scale structure, chemical state, and nature of the intermediate species present on the catalyst surface is necessary, a goal hampered by the heterogeneity of the widely used powder catalysts that make atomic-level investigations difficult.

To resolve this problem, we used two model catalysts[26,27]: one in the form of crystalline nanoparticles (NPs) of $CoMnO_x$, the other in the form of amorphous few-nanometers-thin films of Co, MnO, and $CoMnO_x$, grown by physical vapor deposition on silicon wafers and on silicon nitride ($SiN_x$). An important feature of our catalysts is that in the $CoMnO_x$ NPs and in the thin films the atomic components are mixed at the sub-nanometer (sub-nm) scale, which we verify by TEM and X-Ray diffraction, rather than segregate in different domains sharing boundaries when in contact. Our model catalysts facilitate microscopic and spectroscopic studies under reaction conditions, which reveal the structure of the active sites and the special role of Mn. Our spectroscopic techniques include Ambient Pressure X-ray Photoelectron Spectroscopy (APXPS), and X-ray Absorption Spectroscopy (XAS), in the presence of syngas reactants. The results show that the presence of Mn substantially enhances the formation of $CH_x$ species, which explains the high selectivity for long-chain hydrocarbon products. With DFT calculations we explain the role of $MnO_x$

in controlling the chemical potential of H, which binds to basic O sites of MnO and thus reducing the supply needed for chain termination reactions, while the increase in the concentration of $CH_x$ species favors C-C coupling and chain growth, in agreement with Weckhuysen et al.[10].

## Results and discussion

### Synthesis and characterization of $CoMnO_x$ model catalysts

Crystalline nanoparticles of mixed Co and Mn oxide ($CoMnO_x$ NPs) were synthesized by hot injection of $Co_2(CO)_8$ and $Mn_2(CO)_{10}$ into a solution of octyl-ether in the presence of oleic acid[28–30]. The bulk composition of the NPs was measured by Inductively Coupled Plasma Optical Emission Spectrometry (ICP-OES). High-resolution scanning transmission electron microscopy (STEM) shows that the $CoMnO_x$ NPs are crystals of ~10 nm diameter (Fig. 1a). Two atomic plane distances of 0.248 nm and 0.217 nm are observed, consistent with the (−111) and (002) lattice spacings of CoO (Fig. 1b), indicating that they are solid solutions of Mn in the CoO matrix. The initial oxidation state of CoMn is due to reaction with oleic acid and by exposure to air.

Thin films of $CoMnO_x$ were also prepared, using sequential evaporation of Co and Mn onto a Si wafer and/or onto $SiN_x$ TEM windows. High-angle-annular-dark-field (HAADF), STEM, and energy-dispersive X-ray spectroscopy (EDS) images of $CoMnO_x$ NPs, and of $CoMnO_x$ thin films after the activation process described below, are shown in Fig. 1c–j. The images show that Co and Mn are homogeneously mixed at the sub-nm scale.

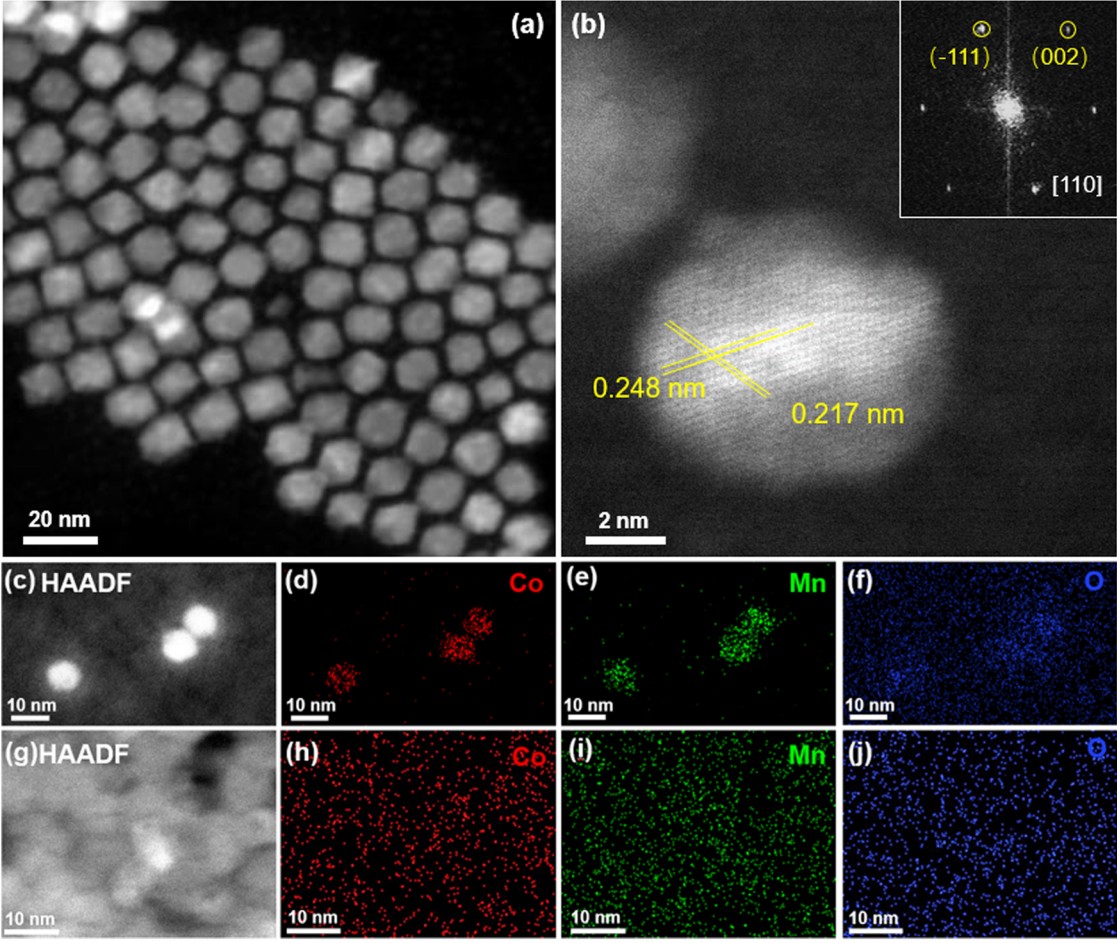

**Fig. 1 | STEM-HAADF and EDS analysis of $CoMnO_x$ NPs and thin film. a** STEM-HAADF image of as-synthesized $CoMnO_x$ crystal NPs; (**b**) High resolution image of a $CoMnO_x$ NP, with an inset showing its FFT pattern; (**c–f**) STEM-HAADF images and EDS maps of $CoMnO_x$ NPs; (**g–j**) STEM-HAADF image and EDS maps of a 10 nm thick amorphous $CoMnO_x$ film grown on a $SiN_x$ membrane. The images (**c–j**) are acquired after reduction by heating to 450 °C in 1 Bar of $H_2$. These images show that in both cases Co and Mn are intimately mixed at the sub-nm scale.

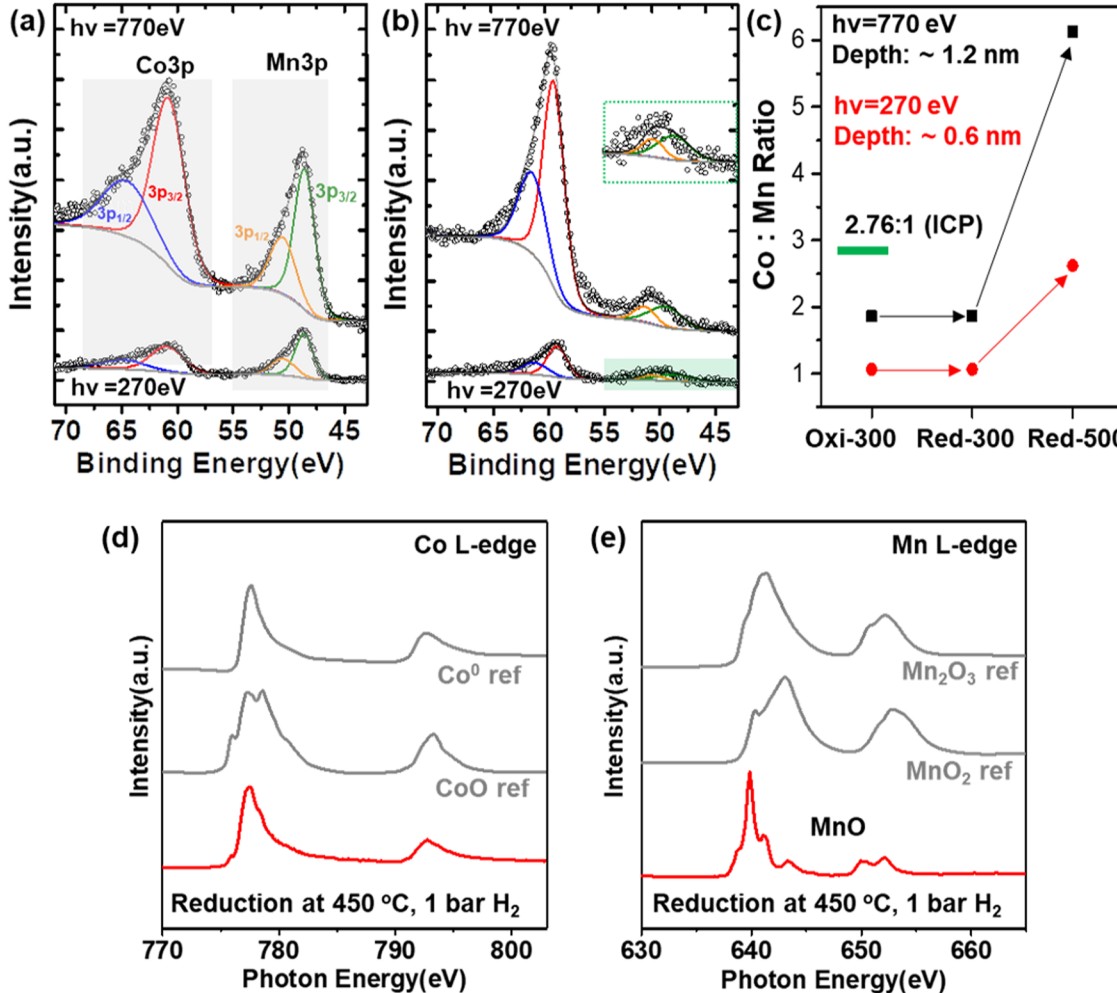

**Fig. 2 | Surface composition of CoMnO$_x$ NPs from Co3p and Mn3p XP spectra. a** under 100 mTorr O$_2$ at 300 °C. **b** under 100 mTorr H$_2$ at 500 °C. Two photon energies, 770 eV and 270 eV were used (top and bottom spectra) to obtain a depth distribution profile based on the different mean-free paths of 1.2 and 0.6 nm respectively of the corresponding photoelectrons. The colored curves are fitting peaks for the 3p$_{1/2}$ and 3p$_{3/2}$ components for Co and Mn. The inset in the green box shows an expanded view of the Mn 3p XPS (hv = 270 eV) region in (**b**). **c** Co:Mn ratio in the top surface region (~0.6 nm, red points) and near surface region (~1.2 nm, black points) after oxidation and reduction treatments. The top surface of the activated catalyst contains Co$^0$ and Mn$^{2+}$ in a 2.5:1 ratio. **d, e** Co and Mn L-edge total electron yield XAS from CoMnO$_x$ NPs (red), after activation by heating in 1 bar of H$_2$ at 450 °C. Spectra from known compounds (gray) are shown for comparison.

In-situ X-ray Diffraction (XRD) patterns of the CoMn NPs, shown in Fig. S1, reveal the CoMn spinel structure after O$_2$ calcination at 300 °C, which produced MnO after reduction in H$_2$ at 450 °C, suggesting segregation of Co from the bulk to the surface region, consistent with the in-situ XPS results discussed below.

**Catalytic activity measurements**

The FTS activity of the activated CoMnO$_x$ NPs model catalysts was measured using a fixed-bed reactor described in the Methods section. Product distributions are shown in Fig. S2 for two Co/Mn composition ratios (left), and for pure Co on Al$_2$O$_3$ (right), along with the prediction from the Anderson−Schulz−Flory (ASF) model for a chain growth probability α of 0.5[1]. Under steady state of 10% CO conversion, 220 °C, 1 bar, and H$_2$/CO ratio of 2, the CoMnO$_x$ NPs catalysts with a Co:Mn ratio between 1.6 and 2.8 displayed higher selectivity for production of hydrocarbons with chain lengths ≥5 carbon units (C$_{5+}$: 48%, CH$_4$: 18%), than Co/Al$_2$O$_3$ catalyst (C$_{5+}$: 25%, CH$_4$: 38%). The product distribution on these CoMnO$_x$ NPs obeys the ASF distribution with α = 0.76, in agreement with previous results[6,23]. These results show that our CoMnO$_x$ NPs model catalysts perform similarly to industrial powder-form catalysts in terms of product distribution, with enhanced percentage of hydrocarbon chains lengths of more than 5 carbons units. Similar catalytic activity measurements could not be carried out on the CoMnO$_x$ films due to the large difference in catalyst area exposed to reactants, which was nearly 4 orders of magnitude higher for the NPs than for the films. However, in spite of this difference, as we show in the following sections, that the elementary steps of the FTS reaction are the same in the thin films and in the nanoparticles.

**Catalyst surface structure after activation**

APXPS characterization results from CoMnO$_x$ NPs after activation, consisting of heating at 300 °C in 100 mTorr O$_2$, to remove contaminants, and subsequent heating to 500 °C in 100 mTorr H$_2$ are shown in Fig. 2. The oxidation process resulted in a mixture of Co$^{2+}$, Co$^{3+}$, Mn$^{2+}$, and Mn$^{3+}$ states as in Co$_3$O$_4$ and Mn$_3$O$_4$ respectively, as shown by comparing the XAS results of our catalyst with standard compounds (Fig. S3a, b). The reduction was carried out under 1 Bar of H$_2$, but the APXPS experiments were carried out in the vacuum chamber where the H$_2$ pressure was 100 mTorr. The structure of the activated CoMn was the same in both. A composition depth profile of the surface region was obtained from the intensities of the Co 3p and Mn 3p XPS peaks acquired at photon energies of 770 eV and

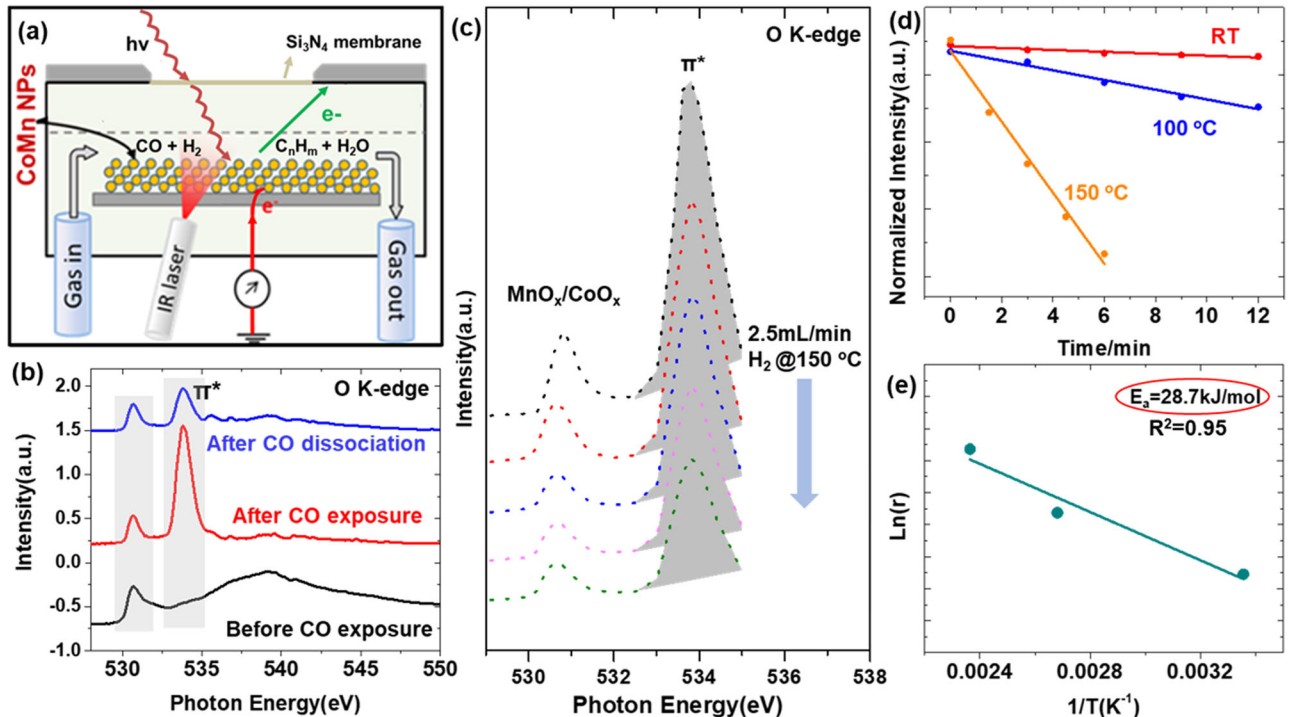

**Fig. 3 | Reactions between CO and H₂ on CoMnOₓ crystal NPs. a** Schematic illustration of the reaction cell used for operando reaction studies using TEY-XAS. The yellow circles represent CoMnOₓ NPs on a gold foil covering a Si wafer. **b** O K-edge XAS of CoMnOₓ before (black), after CO exposure (red), and after H₂ introduction (blue). The peak a 531 eV is due to lattice O in CoMnOₓ. The peak at 534 eV corresponds to the X-rays excitation of electrons from the O 1s level of CO to the antibonding orbital (π*). This peak has contributions from the molecular CO adsorbed on Co⁰, and from carbonates on CoMnOₓ. The broad peak around 540 eV

is due to transitions to unoccupied σ* bands[35], which disappear due to adsorbed CO donating electrons to these orbitals. After introduction of H₂, the peak at 534 eV decreases due to H₂-assisted CO dissociation, leaving only contributions from the carbonate species. **c** Time evolution (2 min intervals) of the CO π* peak intensity under 1 bar of H₂ flowing at 2.5 mL/min at 150 °C. **d** Intensity of the CO π* orbital peak as function of time for three temperatures, which measures the H₂-CO reaction rate. **e** Arrhenius plot from the π* orbital peak intensity decay rate.

270 eV (top and bottom spectra in Fig. 2a, b), which generate photoelectrons of ~700 eV and ~200 eV energies with mean free paths of ~1.2 nm and ~0.6 nm, respectively. We refer to these as 'near surface' and 'top surface' regions. The red/green traces for the 3p₃/₂, and blue/orange traces for the 3p₁/₂ indicate their spin-orbit components. For CoMnOₓ NPs with Co:Mn ratio of 2.76:1 (from ICP-OES), the peak intensities after oxidation indicate a Co:Mn ratio of 1:1 for the top surface region, and 2:1 for the near surface region, indicating surface enrichment of Mn. The reduction under 100 mTorr of H₂ at 300 °C brought no appreciable changes in surface composition (Fig. S4a), and only chemical changes of Co and Mn, which were both reduced to the 2+ state (Fig. S4b, c). However, when the temperature was raised to 450 °C and higher, the surface composition changed significantly, as indicated by the increase of Co 3p XPS peak intensity (Fig. 2b), which now corresponds to a Co/Mn ratio of 6 in the near surface region and 2.5 in the top surface region, respectively (Fig. 2c). The higher temperature reduction changed the oxidation state of Co from 2+ to metallic, while Mn remained in the 2+ state, as can be seen in the XAS of the samples, along with reference spectra from Co, CoO, Mn₂O₃, and MnO₂ (Fig. 2d, e). STEM- EDS maps demonstrate the preservation of the CoMnOₓ bulk structure during oxidation (Fig. S3c) and reduction pretreatment (Fig. 1d–f, h–j), indicating that the pretreatments mostly affect the redistribution of Co and Mn in the surface region. The surface composition of reduced CoMnOₓ was also confirmed by the C1s APXPS results using CO as probe molecule which adsorbs molecularly only on Co⁰ sites and becomes carbonate on MnO sites (Fig. S5). The two-step reduction pattern is similar to that reported for CoMnOₓ catalysts prepared by co-impregnation[8,31].

## Reactions between CO and H₂ on CoMnOₓ NPs, CoMnOₓ, MnO, and Co films

To study the reaction between CO and H₂ on CoMnOₓ, we used a reaction cell like that shown schematically in Fig. 3a. The cell is closed by a 100 nm thick Si₃N₄ window that separates the volume inside, filled with reaction gases, from the beamline vacuum chamber. The CoMnOₓ NPs, were deposited on a gold foil on top of a Si wafer. They were first reduced by exposure to 1 bar of a 10% H₂/Ar mixture at 450 °C, heated by an 808 nm IR photodiode laser via an optical fiber from the back, and then cooled to reaction temperature[28,32]. The NPs chemical composition was determined by XAS recorded in the Total Electron Yield (TEY) mode by measuring the sample to ground current[28], which is surface sensitive to a depth of few nm[33,34]. The O K-edge XAS of the activated sample before CO introduction, is shown by the bottom black curve in Fig. 3b. The peak at ~531 eV corresponds to 1s-level electrons from lattice O in CoMnOₓ excited to states at the bottom of the oxide conduction band[28,35], and the broad peak near 540 eV to excitation to higher unoccupied σ* bands of CoMnOₓ[35]. After exposure to CO, which adsorbs molecularly only on Co⁰ sites, an intense peak at ~534 eV is observed, corresponding to the excitation of O 1s-level electrons of CO to the empty π* antibonding orbital of the molecule[28] (Fig. 3b, red curve). On oxide sites, however, CO adsorbs forming carbonates, which also contribute to the 534 eV peak[35]. After flowing H₂ at 2.5 mL/min at 1 Bar pressure for 12 min with the sample at 150 °C, the intensity of the peak at ~534 eV dropped significantly as a function of time (Fig. 3c) due to the H₂-assisted CO dissociation reaction[28]. The remaining peak, after all molecular CO has dissociated (Fig. 3b, blue curve), corresponds to unreacted carbonates on CoMnOₓ[35]. Figure 3d shows the evolution of the π* peak intensity as a function of reaction

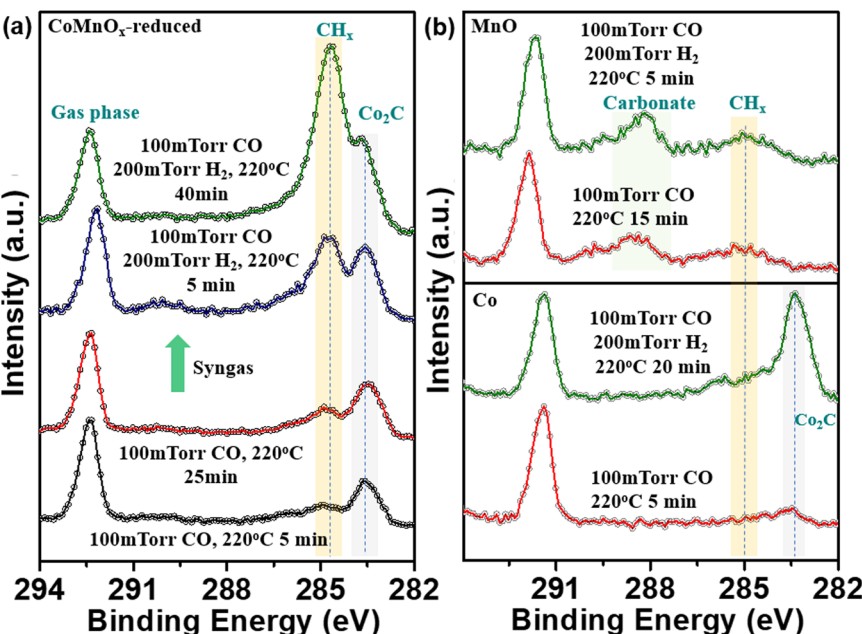

**Fig. 4 | XPS showing surface carbonaceous species on CoMnO$_x$, Co, and MnO thin films. a** From the bottom: activated CoMnO$_x$ under 100 mTorr CO at 220 °C for 5 min (black), after 25 min (red), and under syngas (100 mTorr CO + 200 mTorr H$_2$) for 5 min (blue) and for 40 min (green). The absence of a peak at ~286.0 eV from adsorbed CO is due to its low coverage at 220 °C resulting from desorption/ equilibrium with the gas phase, and from dissociation by reaction with H$_2$.

Most noticeable is the increase in intensity of the peak from CH$_x$ species at ~285 eV. **b** Bottom: metallic Co film exposed to 100 mTorr CO at 220 °C for 5 min (red), and to syngas (100 mTorr CO and 200 mTorr H$_2$) for 20 min (green). CO dissociation by H$_2$ produces cobalt carbide (~283.5 eV) on Co but no stable CH$_x$. Top: pure MnO film under 100mTorr CO at 220 °C for 15 min (red) and under syngas for 5 min (green) showing a smaller peak at ~285 eV, likely due to adventitious contamination.

time for 3 temperatures: RT, 100 °C, and 150 °C. The linear relationship indicates a constant reaction rate. An Arrhenius plot of the rate, Fig. 3e, gives an activation energy of ~28 kJ/mol, a value substantially smaller than that of the typical FTS reaction (~80 kJ/mol)[6], showing a reduction of the barrier for H$_2$ induced CO dissociation on the Co-MnO surface. Besides, the H$_2$-assited carbonate dissociation reaction is unlikely to occur. According to the DFT simulations, the adsorption of CO as carbonate is much less likely than molecular adsorption on the Co phase of the catalyst (−0.28 eV versus −1.66 eV). Therefore, these carbonates adsorbed on the oxidic part are likely spectators, as any catalytic cycle involve carbonate will require the activation from the metallic Co phase with an energetic cost of at least 1.38 eV. It agrees with the in-situ O 1s XAS results as after the H$_2$ introducing, the unreacted carbonates still remain on CoMnO$_x$ catalyst surface.

In addition to XAS, the chemical state of the CoMnO$_x$ NPs under FTS reaction conditions was followed by APXPS. After reduction by exposure to 100 mTorr H$_2$ at 500 °C (Fig. S6a, black curve), the spectra show that most of the Co atoms in the top surface region (~4.6 Å), are in the metallic state, while within the same depth, (Fig. S6b), Mn remains in the 2+ state. Following exposure of the reduced sample to 100 mTorr of CO at 220 °C, a fraction of the metallic cobalt was oxidized to CoO due to CO dissociation[28], as indicated by the increased shoulder intensity at ~780.2 eV from Co$^{2+}$ (Fig. S6a, red curve). Under 300 mTorr of syngas at a ratio CO:H$_2$ = 1:2, the oxide peak increased substantially and becomes the dominant peak (Fig. S6a, blue curve). The increased oxidation is the result of H$_2$-assisted dissociation of CO, confirmed by Co$^{2+}$ peak in the Co L-edge XAS (Fig. S7a). During this period, no significant chemical change in the MnO was observed (Figs. S6b, S7b). More important, a depth profile using the XPS peaks of Co 3p and Mn 3p (Fig. S7c, d) demonstrates that under syngas reaction conditions at 220 °C, the composition of top and near surface regions remains unchanged during reaction, indicating the stability of the Co-MnO structure. It also shows that oxidized Co, from H$_2$-assisted CO dissociation, is subsequently reduced by H$_2$. This structural stability

under the FTS reaction conditions was further confirmed by in-situ XRD in Fig. S1 and ex-situ TEM in Fig. S8, which demonstrates that although some morphological changes occur, the Co and Mn atoms remain intimately mixed at the sub-nm scale.

The nature of the species formed on the catalyst surface during the reaction of H$_2$ with CO on each Co, MnO and CoMnO$_x$ films is further revealed by APXPS in the C 1s region, shown in Fig. 4. Starting with a 100 mTorr of CO gas environment, when the temperature was raised to 220 °C, the carbonates at ~289.0 and ~289.5 eV desorbed from on MnO and CoMnO$_x$ (Fig. S5), while a peak at ~283.5 eV from Co carbide increases noticeably on Co and somewhat less on CoMnO$_x$, where there is an apparent increase due to its overlap with the ~285 eV peak from CH$_x$ species. These C atoms for the reactions originate from H$_2$-assisted CO dissociation[28]. The most remarkable change is the growth of the carbon peak at ~285 eV due to CH$_x$ species, which occurs only on CoMnO$_x$ but not on Co or MnO. There is however a small contribution to that peak in Co and MnO from contaminant species from background gases[36–40] that produce species with peaks between 284–285 eV. After 40 min of reaction, top green curve in Fig. 4a, the CH$_x$ peak is the dominant and most stable carbonaceous species on the surface, as shown by the higher temperature, above 400 °C, needed to desorb it, as shown in the sequential spectra acquired versus time (Fig. S9). The increase in concentration, and higher thermal stability of the CH$_x$ species indicates that it belongs to the longer chain intermediates formed by the chain-growth reaction[41,42], because longer chain molecules bind more strongly to the surface than shorter ones, with methane binding the weakest. Highly significant is also the observation that under the same conditions, no significant growth of CH$_x$ species is observed on pure metallic Co or on MnO (Fig. 4b).

## Theoretical simulations

To better understand the molecular scale origin of the higher selectivity of CoMnO$_x$ catalysts towards C$_{5+}$ hydrocarbon products in the FTS reaction, we performed DFT simulations on a model catalyst with a

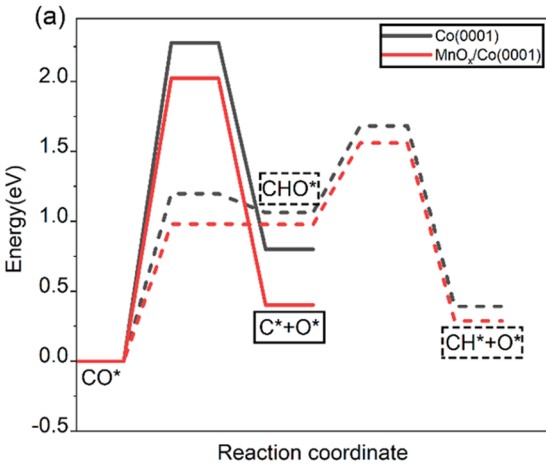

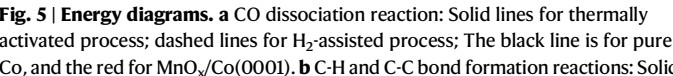

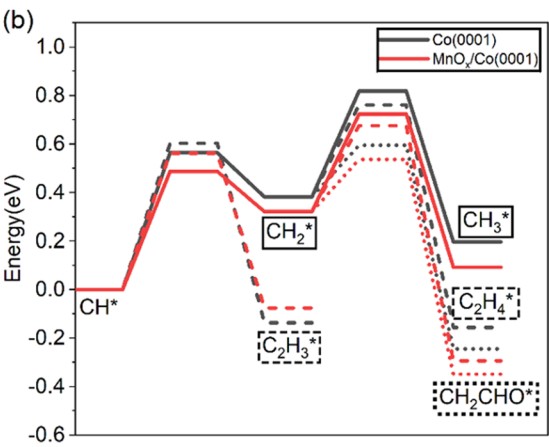

**Fig. 5 | Energy diagrams. a** CO dissociation reaction: Solid lines for thermally activated process; dashed lines for H$_2$-assisted process; The black line is for pure Co, and the red for MnO$_x$/Co(0001). **b** C-H and C-C bond formation reactions: Solid lines for CH* + H* reactions. Dashed lines for CH$_x$*+CH$_2$* reactions. Dotted lines for CH$_2$* + CHO* reaction. Relative energies between elementary steps were shifted to align the initial state for energy barrier comparisons.

structure and composition based on the experimental data described above: a Co:Mn ratio of ~2.5 in the topmost layer with Co in the metallic state (Co$^0$), and Mn with a double positive charge (Mn$^{2+}$). A model that fits this ratio and oxidation state is one where the components Co, Mn, and O are arranged in a compact structure. And interfacial structure was determined by combining global optimization (See SI computational details) and the calculation of stability under reaction conditions (Fig. S10), as that illustrated in Fig. S10a (center) with a unit cell in the topmost layer containing 12 Co atoms, 4 Mn atoms and 4 O atoms. A Bader charge analysis shows that the Mn in the model has a charge of +1.20 |e|, compared with that in MnO (+1.47 |e| for Mn in the bulk, and +1.39 |e| for Mn on the surface). For comparison the Mn charge in Mn$_2$O is +0.72 |e| for Mn in the bulk. The average charge of surface Co atoms is +0.00 |e|, and the average charge of the Co atoms near MnO$_x$ is +0.07 |e|, indicating that Co is mostly metallic but slightly polarized near the MnO interface. The projected density of states (PDOS) of the Co d-band in Fig. S11, shows only a small difference between that from Co atoms in Co(0001) and in CoMnO$_x$. This structure is stable against changes in oxidation state by either gain or loss of oxygen atoms illustrated in Fig. S10a–c, respectively.

The adsorption energies of relevant key fragments were then calculated on three simple surfaces (Table S1): MnO(100), metal Co(0001), and our model MnO$_x$/Co(0001). The adsorption of H*, CO*, CH*, and CH$_2$* on MnO(100) was found to be weaker than on Co(0001) by 1.26 eV, 1.64 eV, 2.54 eV, and 1.86 eV, indicating that the species prefer to adsorb on Co instead of MnO. In the MnO$_x$/Co(0001) model, 6 sites near the interface are considered, as shown in Fig. S10a, which are top sites (marked by circles), hcp and fcc 3-fold sites (marked by down-pointing and up-pointing triangles). On MnO$_x$/Co(0001), most of the adsorbates bind preferentially to the Co sites instead of MnO sites with an adsorption energy only slightly different from that in pure Co(0001). Some carbon fragments (CH* and C$_2$H$_4$*) do not adsorb on MnO$_x$ sites of the MnO$_x$/Co(0001) model, while for CO* and CH$_2$*, the adsorption is more endothermic than on the Co sites by 1.60 eV and 1.45 eV. For H*, O*, and C* the adsorption is only moderately more endothermic than on the Co sites by 0.15 eV, 0.37 eV, and 0.25 eV, respectively. The diffusion of H from Co to O sites on MnO$_x$/Co(0001) was found to be sensitive to the local interface structure when the oxygen content changes. For example, the energy barrier of H diffusion from Co to O sites near the interface changes from 1.47 to 0.85 eV (Fig. S12), indicating that the diffusion is easier on configurations with certain local oxygen environments.

DFT calculated energy diagrams for the reaction steps are shown in Fig. 5. In panel (a) the continuous lines correspond to thermal CO dissociation, while the broken lines are for H$_2$-assisted dissociation. The continuous lines are for reactions on Co sites in Co(0001) (black), and for Co sites in the MnO$_x$/Co(0001) surface (red). The energy barrier of the rate determining step (RDS) for H$_2$-assisted dissociation on Co(0001) is 1.20 eV, while the energy barrier of CO thermal dissociation is 2.28 eV. On MnO$_x$/Co(0001), both energy barriers are decreased, the former to 0.98 eV, and the latter to 2.03 eV.

A charge density difference analysis shows that CO has a stronger interaction with Co on the MnO$_x$/Co(0001) surface than with Co on the Co(0001) surface (Fig. S13). Assuming the pre-exponential factor of the reaction rate to be the same for the Co sites on Co(0001) and Co sites on MnO$_x$/Co(0001), the reaction rate on MnO$_x$/Co(0001) is two orders of magnitude higher than on pure Co(0001) at ~220 °C. Therefore, the presence of the interface will facilitate CO dissociation and increase CHO* formation, which further decomposes to CH*.

Reactions involving carbon fragments can proceed in two different ways (Fig. 5b): (i) oligomerization, i.e., C-C coupling, and chain growth, or (ii) termination by hydrogenation of the carbon moieties. The competition between these two processes leads to the ASF distribution[43]. The hydrogenation of CH* and CH$_2$*, and the coupling of CH* + CH$_2$*, CH$_2$* + CH$_2$*, CH$_2$* + CHO*, reactions were calculated to understand the effect of Mn in the formation of C-H and C-C bonds. The CH$_2$* + CHO* coupling reaction has an energy barrier of 0.21 eV on Co(0001), and 0.22 eV on MnO$_x$/Co(0001), indicating that C-C coupling can occur through the reaction, depending on the relative population of both species. Since CHO* easily dissociates into CO* and H*, the coverage of the intermediate CHO* will be low, leading to a low rate of this bimolecular reaction, and for this reason the reaction is not discussed further in the following. In general, the C-C formation releases more energy than the C-H formation, indicating that C-C bond formation is thermodynamically favorable (Fig. 5b, Table S2). Starting from CH* fragments, the energy barrier towards C-H formation (0.57 eV on Co, 0.49 eV on MnO$_x$/Co(0001)) is lower than the C-C formation (0.60 eV on Co(0001), 0.56 eV on MnO$_x$/Co(0001)). However, once CH$_2$* is formed, the energy barrier for C-C coupling (0.38 eV on Co, 0.36 eV on MnO$_x$/Co(0001)) is lower than C-H formation (0.44 eV on Co(0001), 0.40 eV on MnO$_x$/Co(0001)), indicating that C-C formation is likely to occur with CH$_2$*.

From the simulations, the following picture of the reaction emerges (Fig. 6) that explains the impressive performance of CoMnO$_x$. First, syngas species adsorb on the metal and H$_2$ dissociates, with some

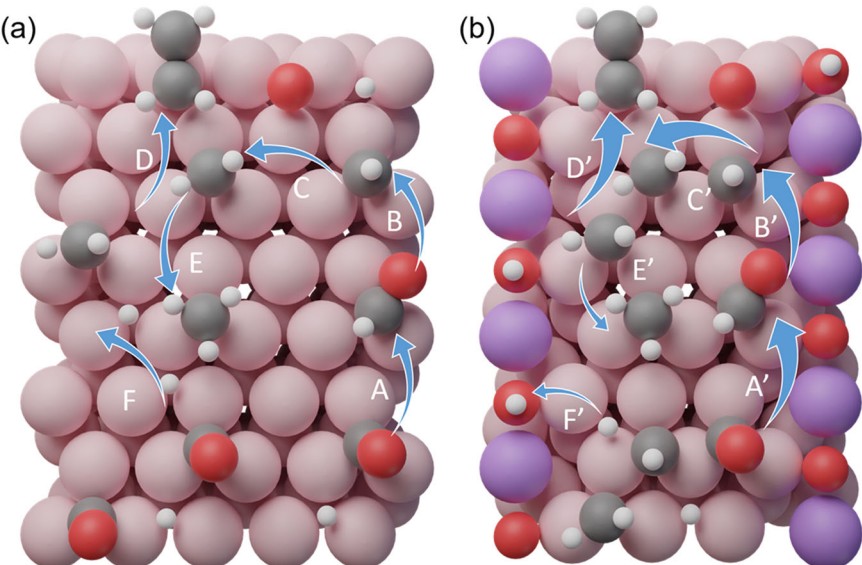

**Fig. 6 | Illustration of the proposed reaction mechanism. a** on Co(0001) and (**b**) on CoMnO$_x$. Co atoms shown in pink, Mn in purple, O in red, H in white, and C in gray. The arrows' size follows qualitatively the value of the reaction rate for comparison on different catalysts. A more detailed illustration of the reaction steps is shown in Fig. S16.

H atoms being captured on O sites in MnO$_x$ to form hydroxyls species (F' in panel b), in agreement with the appearance of a shoulder in the XPS peak of O 1s at ~531 eV (Fig. S14). Hydroxyls from the Cobalt surface may also contribute to this peak[44]. Once the H atoms bind to basic centers (O) of the CoMnO$_x$ system, they are sequestered and unavailable for C-H formation (E') (Figs. 6 and S15). The interface shows good activity in activating CO via the H$_2$-assisted pathway (A'-B'), which increases the rate and overall activity of formation of CH$_x$* species (C'). Second, chain growth steps (D') are more likely on CoMnO$_x$ compared to pure Co, from both coverage and energetic considerations: CH$_x$* adsorption on MnO is unfavorable, therefore facilitating the increase of CH$_x$* coverage on Co. In addition, the barrier for C-C coupling is lower on CoMnO$_x$ than that on pure Co, which favors chain growth. The step-termination reactions are less likely on CoMnO$_x$ because a fraction of H atoms are bound to the oxide and some consumed to remove some of the O generated by CO dissociation. Thus, the MnO$_x$ acts as a buffer for the chemical potential of both oxygen and H under reaction conditions making them optimal to reduce side reactions.

It has been proposed in recent literature that Co$_2$C, in the form of nanoprisms exposing (101) and (020) facets, are active for selective olefin conversion[18]. However, the CoMnO$_x$ NP crystal catalysts used here mainly produces C$_{5+}$ hydrocarbons, indicating that the Co$_2$C, present in small amounts in pure Co and even less in CoMnO$_x$, as shown by the C 1s APXPS in Figs. 4 and S5, has a negligible contribution to the FTS products in our case. To explain this we performed DFT calculations using the stoichiometric surface Co$_2$C(011) surface, which has the lowest surface energy[45]. The results, shown in Fig. S17, indicate that for all the pathways shown in Fig. 6, the energy barriers on Co$_2$C(011) are higher than on Co(0001).

In summary, by using well-defined CoMnO$_x$ crystalline nanoparticles and amorphous thin films as model catalysts where Co and Mn are not separated but mixed at the sub-nm scale, together with in-situ spectroscopic characterization by APXPS and XAS under operating conditions, and with the help of DFT calculations, we unraveled the elemental distribution of Co and Mn atoms on the surface of the catalyst and their oxidation state, showing also that the bulk structure remains practically unchanged. The catalytically active phase of CoMnO$_x$ under FTS reaction conditions is composed of metallic Co in

contact with MnO. The reaction proceeds by H$_2$-assisted CO dissociation and leads to a large increase in the production of CH$_x$* intermediates, much more than in pure Co, favoring chain growth reactions. The main advantage of CoMnO$_x$ is that Mn remains always oxidized with its O providing basic sites that bind H, thus lowering the amount available for CH$_x$*+H* coupling, leading to chain termination and methane formation, while increasing the concentration of CH$_x$* species, which favors the C-C coupling between towards oligomerization and chain growth.

## Methods

### Preparation of model catalysts

The CoMnO$_x$ NPs were synthesized as follows: a solution of 0.5 mL of oleic acid in 15 mL of dioctyl ether was stirred under vacuum and heated to 50 C for 30 min. Next, a mixture of 205 mg of Co$_2$(CO)$_8$ and 117 mg of Mn$_2$(CO)$_{10}$ (corresponding to a feeding ratio of Co:Mn = 2:1) in 2 mL of dioctyl ether was injected in the solution. The solution was heated at 5–8 °C /min and kept at 290 °C for 1 h, followed by cooling down to room temperature, with isopropanol added to precipitate the nanoparticles. The nanoparticles were purified by 2 cycles of centrifugation and dispersed in hexane. This synthesis produced CoMn nanocrystals with diameters of ~10 nm and Co:Mn composition ratio of 2.76:1, as measured by Inductively Coupled Plasma Optical Emission Spectrometry (ICP-OES).

CoMnO$_x$ thin film were prepared by sequential evaporation of Co and Mn onto a Si wafer and onto SiN$_x$ TEM windows and heated to 300 °C under 1 bar O$_2$ atmosphere. Their thickness, determined from Quartz Crystal Microbalance (QCM) measurements, was 100 nm for the Si wafer substrates and 10 nm for the silicon nitride TEM windows, respectively. Other two film compositions, Co and Mn, were prepared side-to-side on the same Si wafer. In this manner comparison of their structure under the same conditions, and under the same oxidation/reduction procedures as those used for the CoMnO$_x$ NPs, was possible.

### TEM characterization

The as-prepared CoMnO$_x$ NPs were drop-cast onto a SiN$_x$ TEM window (10 nm thickness, VWR Scientific). High resolution TEM studies were performed in an aberration corrected FEI Titan 80300 operated at

300 kV and equipped with a CEOS GmbH double hexapole aberration corrector providing angstrom level resolution in scanning imaging modes. Prior to TEM characterization, the CoMnO$_x$ NPs underwent activation, consisting of two steps: first removal of the carbon by heating in a gas flow of 20% O$_2$/Ar mixture, at 25 mL/min, 1 bar, at 300 °C for 2 h. The second step is a reduction in a flow of 20% or 10% H$_2$/He mixture at 25 mL/min, 1 bar, for 2 h. The reduction step was done at two different temperatures of 300 °C and 450–500 °C.

## FTS catalytic activity measurements

The FTS catalytic activity of the CoMnO$_x$ NPs was tested using 70 mg of NP particles supported on alumina powder by sonication with a loading of 10 wt%. The fixed-bed reactor used was heated to 450 °C with a ramp of 5 °C /min for 2 h in flowing H$_2$ and subsequently cooled to 220 °C in flowing He. After activation the gas flow or was gradually switched from He to syngas over 10 min. The syngas was composed of H$_2$, CO (H$_2$/CO at a ratio of 2:1), and 7% Ar (Praxair, 99.999% purity) used as an internal standard. The catalyst temperature was measured using a K-type thermocouple positioned in the bed center. An Agilent 6890 N gas chromatograph was used to monitor the chemical composition of the reactor effluent. The gas chromatograph was equipped with a capillary column connected to a flame ionization detector to measure the hydrocarbon products, and a packed column connected to a thermal conductivity detector for measuring H$_2$, Ar, and CO.

## In-situ APXPS and TEY-XAS characterization

For in-situ APXPS and TEY-XAS experiments, the CoMnO$_x$ NPs were drop-cast onto an Au foil and activated as indicated above. The APXPS results were acquired in beamline 9.3.2 of the Advanced Light Source (ALS) of the Lawrence Berkeley National Laboratory (LBNL). TEY-XAS experiments were performed using a home-built gas cell in beamline 8.0.1.4 of the ALS with an energy resolution of 0.2 eV for the Co L-edge, Mn L-edge and O K-edge. The reaction cell is closed by a 100 nm thick Si$_3$N$_4$ window that separates the volume inside, filled with 1 Bar of reaction gases from the high vacuum chamber[28,32]. The CoMnO$_x$ NPs, supported on the Au foil, were heated using an 805 nm IR laser via an optical fiber that illuminates the sample from the back. Copper gas lines were used to introduce the CO and H$_2$ reactants with Cu carbonyl traps kept at 240 °C to remove any carbonyls present. All measurements were conducted under flowing gas conditions.

## Data availability

The model system, structures of adsorption and reaction are available in ioChem-BD database[46] (https://doi.org/10.19061/iochem-bd-6-272).

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

## Acknowledgements

This work was supported by the Office of Basic Energy Sciences (BES), Chemical Sciences, Geosciences, and Biosciences Division, of the U.S. Department of Energy (DOE) under Contract DE-AC02-05CH11231, FWP CH030201 (Catalysis Research Program). It used resources of the Advanced Light Source, a U.S. DOE Office of Science User Facility under contract no. DE-AC02-05CH11231, and at the 23-ID-2 (IOS) beamline of the National Synchrotron Light Source II, a User Facility operated for the DOE Office of Science by Brookhaven National Laboratory under Contract No. DE-SC0012704, and at the Stanford Synchrotron Radiation Light Source, SLAC National Accelerator Laboratory, supported by the U.S. Department of Energy, Office of Science, Office of Basic Energy Sciences under Contract No. DE-AC02-76SF00515. X.Z. was supported by an NSF-BSF grant number 1906014. J.W. and H.Z. acknowledge the support of the U.S. Department of Energy, Office of Science, Office of Basic Energy Sciences (BES), Materials Sciences and Engineering Division under Contract No. DE-AC02-05-CH11231 within the in-situ TEM program (KC22ZH). N.L and Z.L. acknowledge the support from the European Union Horizon research and innovation program under the Marie Skłodowska-Curie grant agreement No 101064867, and the Spanish Ministry of Science and Innovation (PID2021-122516OB-I00, Severo Ochoa Center of Excellence CEX2019-000925-S 10.13039/501100011033). The authors also thank the Barcelona Supercomputing Center (BSC-RES) for providing the computation resources. H.C. thanks X.Z., E.W., Dr. Carlos Escudero (ALBA Synchrotron, Spain) and Dr. Sergio López Rodríguez (University of Alicante, Spain) for the troubleshooting of the home-built in-situ XAS gas cell. H.C. thanks Dr. Terry McAfee (1987–2024), a Senior Scientific Engineering Associate at ALS(LBNL), for his great support in the APXPS experiment at ALS Beamline 9.3.2. H.C. thanks David Malone (ALS, LBNL), Alyssa Brand (ALS, LBNL) and Greta Toncheva (Laser Safety Officer, LBNL) for the gas cell setup and safety inspection. Work at the Molecular Foundry was supported by the Office of Science, Office of Basic Energy Sciences, of the U.S. Department of Energy under Contract No. DE-AC02-05CH11231.

## Author contributions

M.S. oversaw the project. H.C. and X.Z conduct the in-situ APXPS/XAS/XRD experiment supported by the beamline scientists: M.B., A.H., I.W., H.O., J.G., and S.H.L. Data analysis were done by H.C. and M.S. Simulations were carried out by Z.L. and N.L. Sample preparation, characterization and reactivity measurement were done by H.C., P.F.P., X.Z., J.W., J.O.M, J.Y., J.S., E.P., S.C., L.T., N.L., Y.S., Z.Z., F.Y., E.W., and V.A. The paper was written by H.C., Z.L., N.L., and M.S., with contributions from P.Y. and H.Z. and other co-authors. A.T.B. helped to review and edit the manuscript.

## Competing interests

The authors declare no competing interests.
