## [Transparent Peer Review file · Nature Communications]

The Role of Manganese in CoMnOx Catalysts for Selective Long-Chain Hydrocarbon Production via Fischer-Tropsch Synthesis

Corresponding Author: Professor Miquel Salmeron

Version 0:

Reviewer comments:

Reviewer #1

(Remarks to the Author)

FTS is a very important catalytic reaction in both scientific research and industrial application. CoMn catalysts exhibit very unique properties in that it can produce both light olefins and long-chain hydrocarbons. Understanding the promotion effect of Mn on Co-based catalysts is an important scientific problem. The manuscript of Chen et al. studied the origin of the enhancement of Mn promotion in molecular scale by various advanced in-situ characterization methods and theoretical study. It was found that the concentration of CH_x intermediate species increases rapidly under FTS reaction conditions on CoMnO_x, while no increase occurs in the absence of Mn. DFT revealed that the basic O sites in MnO bind Hydrogen atoms generated by H₂ dissociation on the Co sites, making them less available to react with CH_x intermediates, and thus hindering chain termination reactions that produce short-chain hydrocarbons. Although many advanced characterizations have been performed, the conclusion is not novel. The authors should explain what new insights have been found compared to previous work (such as, 10.1021/jp0403846, 10.1016/j.jcat.2004.11.047, 10.1016/j.jcat.2012.01.008, 10.1021/acscatal.5b01578, and so on). In addition, there are several issues to be further discussed.

1. The authors should explain the purpose of preparing CoMnO_x thin films sample so that the readers can understand it better. According to Figure 1h and 1i, the elementary composition and distribution of NPs and thin films samples differ greatly. Additionally, the catalytic performance of thin films is absence. Whether it is appropriate to use this sample for subsequent characterization.
2. There are two CoMnO_x NPs catalysts with a Co:Mn ratio between 1.6 and 2.8. Why the authors choose the latter to conduct subsequent characterization.
3. In Figure 3, the authors calculate the activation energy of H₂-assisted CO dissociation based on the assumption that the hydrogenation of carbonate species does not occur. Please provide more evidence to prove that only H₂-assisted CO dissociation happens.
4. In Figure 4a, the signal at 284.8 eV is assigned to CH_x species. The authors should consider the contribution of carbon deposition.
5. The authors state that H atoms being captured on O sites in MnO_x to form hydroxyls species, in agreement with the appearance of a shoulder in the XPS peak of O1s at ~531eV (Fig.S14). The authors should compare the O1s of Co sample to confirm the produced surface hydroxyl on MnO_x.
6. The authors state that CH_x* adsorption on MnO is unfavorable, therefore facilitating the increase of CH_x* coverage on Co. Please give more explanation. In addition, the energy barriers of CH₂-CH₂ coupling on Co (0.38 eV) and MnO_x/Co (0.36 eV) are too close. This may not reflect the promotion effect of Mn on C-C coupling over Co catalyst.
7. The authors state that H₂-assisted CO dissociation is not the RDS according to Figure 3e. However, DFT results suggested that H₂-assisted dissociation is the RDS. Please give more explanation.

Reviewer #2

(Remarks to the Author)

The manuscript presents a comprehensive study on the role of manganese (Mn) in CoMnOx catalysts for the Fischer-Tropsch synthesis (FTS) reaction. The authors aim to provide insights into the molecular mechanisms underlying the enhanced selectivity for long-chain hydrocarbons on CoMnOx surfaces by combining experimental and theoretical approaches. However, there is a noticeable gap between the theoretical predictions and the experimental observations. Addressing this gap is crucial for enhancing the robustness and reliability of the conclusions drawn from the study. This work employs a relatively simple theoretical model, while the experimental characterization mainly focuses on analyzing the surface chemical composition, lacking structural information. The microstructure has a significant impact on the reaction mechanism and activity. Therefore, the reasonableness of the theoretical calculation model used in this study is worth considering, or further explanation is needed to justify the validity of the theoretical model. Considering that the theoretical calculations occupy a significant portion of this work and their prominent role in the final conclusions, I believe the manuscript does not yet meet the standards for publication. The integration of theoretical calculations and experimental characterization, as well as the justification of the theoretical model's validity—such as its thermodynamic stability and stability under reaction conditions—need to be further strengthened.

Reviewer #3

(Remarks to the Author)

In "The role of Manganese in CoMnOx catalysts in the Fischer-Tropsch reaction to enhance selectivity for long-chain hydrocarbon products" The authors measure the surface composition of working model Co/MnOx catalysts at (very) low pressures of CO and H2.

The Results of the authors imply that paraffinic hydrocarbons are the main product of the reaction in their reaction conditions, which I find surprising, especially with the unusually high Manganese content (approx. 30 % rel. Co). At such high Manganese content, I would expect a high olefin selectivity and even a not negligible amount of oxygenates, at least at more technical reaction pressures (several to tens of bar). The authors address this on p. 15, but only very briefly and, in my opinion, slightly to briefly.

Was the composition measured by ICP-OES or AES? There is a discrepancy between the results and experimental section. The article provides insight to a possible mechanism of the Fisher-Tropsch synthesis -a topic that is still under wide debate- and contributes to the understanding of the reaction.

All in all, I find it is meaningful and well written.

Version 1:

Reviewer comments:

Reviewer #1

(Remarks to the Author)

The authors have made very important changes and improvements in the manuscript that I believe address most of the referees' comments. I recommend acceptance in its current form.

Reviewer #2

(Remarks to the Author)

I appreciate the authors' efforts in responding to my previous comments and the detailed explanation provided regarding the theoretical model and its validation. However, I believe the response fails to adequately address several critical points, and the manuscript still does not meet the necessary standards for publication. Below are my concerns with the authors' reply:

1、 Gap Between Theory and Experiment: The authors claim that their DFT model and experimental results are mutually supportive, yet they fail to sufficiently bridge the gap between theoretical predictions and experimental observations. The theoretical models are built based on several assumptions, and the experimental data presented is insufficient to justify the accuracy and relevance of these models to real-world systems. The authors state that they employed the Co(0001) surface in their calculations based on "(iii) following standard practice in modeling (taking the lowest energy structures) we employ the most compact surfaces for each of these two systems.". While Co(0001) may indeed be the most stable surface, it is widely recognized that Co(0001) is unlikely and/or not necessarily true to serve as the active site for Fischer-Tropsch synthesis. Therefore, comparing reaction activity and selectivity results to Co(0001) remains a great uncertainty. The suitability of constructing the CoMnOx model based on Co(0001) needs to be reconsidered more carefully, not only relying on the authors' own experimental characterization but also incorporating the broader understanding of active sites in cobalt catalysts for FTS from past research. This broader perspective is essential to ensure that the chosen model is representative of actual catalytic behavior in FTS.

2、 Structural Considerations: While the authors considered the structural information from XRD and TEM, there is still a lack of sufficient structural characterization, particularly regarding the microstructure of the catalyst and how it influences the reaction mechanism. The reply provides some justification for the theoretical model, but it does not fully resolve the concerns about the structural complexity of the catalyst, which may play a critical role in its catalytic behavior. The authors claim that the "top surface regions" are approximately 0.6 nm thick, which actually corresponds to 2-3 atomic layers. However, characterization of the "top surface regions" alone is insufficient justification for simplifying the model to this extent. Multiple potential interfacial configurations should have been considered, rather than simply replacing two rows of Co atoms with

MnO on the (0001) surface. Furthermore, the choice of the (0001) surface itself is questionable, as it is not representative of the active sites typically associated with Fischer-Tropsch synthesis. The current model appears somewhat hasty and lacks the thoroughness needed to accurately capture the complexity of the catalyst's structure and behavior. A more detailed exploration of possible interface configurations would be necessary to ensure a robust and representative model.

In conclusion, while the authors have made some effort to clarify their approach, significant issues remain with the integration of experimental and theoretical work. I recommend that the authors strengthen the connection between their theoretical predictions and experimental results, provide more experimental validation for the chosen model, and address the concerns regarding the structural complexity of the catalyst. Without these improvements, the manuscript is not ready for publication.

Reviewer #3

(Remarks to the Author)

The authors seem to have reacted to the critiques and have implemented according changes in the manuscript. In my opinion the manuscript is publishable.

Version 2:

Reviewer comments:

Reviewer #2

(Remarks to the Author)

The authors have provided sufficient explanations and clarifications regarding the issues I was concerned about. I find their responses satisfactory. Therefore, I recommend accepting the manuscript.

Response to the Reviewer 1:

Reviewer #1 (Remarks to the Author):

FTS is a very important catalytic reaction in both scientific research and industrial application. CoMn catalysts exhibit very unique properties in that it can produce both light olefins and long-chain hydrocarbons. Understanding the promotion effect of Mn on Co-based catalysts is an important scientific problem. The manuscript of Chen et al. studied the origin of the enhancement of Mn promotion in molecular scale by various advanced in-situ characterization methods and theoretical study. It was found that the concentration of CH_x intermediate species increases rapidly under FTS reaction conditions on CoMnO_x, while no increase occurs in the absence of Mn. DFT revealed that the basic O sites in MnO bind Hydrogen atoms generated by H₂ dissociation on the Co sites, making them less available to react with CH_x intermediates, and thus hindering chain termination reactions that produce short-chain hydrocarbons. Although many advanced characterizations have been performed, the conclusion is not novel. The authors should explain what new insights have been found compared to previous work (such as, 10.1021/jp0403846, 10.1016/j.jcat.2004.11.047, 10.1016/j.jcat.2012.01.008, 10.1021/acscatal.5b01578, and so on). In addition, there are several issues to be further discussed.

Reply: We thank the Reviewer for the time taken in the assessment of our manuscript.

We agree that the conclusion of our work about the role of Mn in FTS is not new and that it agrees with that of other papers published on the topic. However, while we reach similar conclusions about the role of the Mn promotion, our paper provides new insights and explanation of how Mn promotes the reaction, based on the use of advanced *in situ* and *operando* characterization, combined with a deep theoretical analysis using DFT calculations based on experimentally determined catalyst structures (i.e., atomic distribution at the surface, oxidation state, formation and chemical evolution of intermediate species), obtained by microscopy and spectroscopic techniques, and not on proposed models.

Two clear examples are:

1) The spectroscopic measurement of the reaction products of CO with H₂, and its rate versus temperature. The measurements were carried out *while* (i.e., in operando) the reaction products were being formed, as shown in **Fig.3**.

2) A second example is the formation of the crucial CH_x intermediates, which form and increase in concentration only in catalysts that contain Mn, as shown in **Fig. 4a**, and showing at the same time that that reaction does not occur in the absence of Mn (**Fig.4b**).

Finally, our TEM microscopy results reveal the close interaction between Co, Mn and O due to their close spatial proximity within areas of nm dimensions.

Our comments on the papers mentioned by the Reviewer, and their implication for our work, are described below:

(1) Morales, F.; de Groot, F. M. F.; Glatzel, P.; Kleimenov, E.; Bluhm, H.; Hävecker, M.; Knop-Gericke, A.; Weckhuysen, B. M. *In Situ X-Ray Absorption of Co/Mn/TiO₂ Catalysts for Fischer–Tropsch Synthesis*. *J. Phys. Chem. B* **2004**, *108* (41), 16201–16207. <https://doi.org/10.1021/jp0403846>.

In this paper Prof. Weckhuysen's group studied the reduction reactions of Co, Co/TiO₂ and CoMn/TiO₂ in the presence of **2 mbar** of H₂, to determine the chemical state of Ti, Co and Mn as the temperature increased from RT to 425 °C. In our work we used a home-made gas cell that allowed for *in-situ* XAS measurements of the reaction under **1 bar** of H₂, as shown in our **Fig. 3**. In this manner we could measure the evolution of CO reactants as they dissociate under H₂, and the formation of CH_x species, the two most important steps in the FTS, as a function of temperature. These conditions are closely related to those of industrial reactions.

(2) Morales, F.; de Groot, F. M. F.; Gijzeman, O. L. J.; Mens, A.; Stephan, O.; Weckhuysen, B. M. Mn Promotion Effects in Co/TiO₂ Fischer–Tropsch Catalysts as Investigated by XPS and STEM-EELS. *Journal of Catalysis* **2005**, *230* (2), 301–308. <https://doi.org/10.1016/j.jcat.2004.11.047>.

This paper, also from Prof. Weckhuysen's group, focuses on elucidating the role of Mn promotion on the activity of Co/TiO₂ for the Fischer–Tropsch reaction. They use XPS and STEM-EELS to determine the chemical state of the Co, Mn, and Ti after *H₂ induced reduction*, and the spatial relation between MnO_x and Co active sites. Our XPS results show that after reduction and in the

presence of H₂, Co and Mn are in the 0 and 2+, oxidation states, respectively. The STEM-EELS results in Morales et al. paper show segregation of Co and MnO particles and the existence of an interface between them. However, the spatial resolution is too low and does not reveal the proximity, at the atomic level, of Co, Mn and O. However, we agree with their conclusion: “*the Co–Mn interaction gives rise to the observed promotion effect in FTS. This promotion effect becomes apparent as a suppression of CH₄ formation, an increase in the C₅₊ selectivity, and an enhancement of the Co-time yield*” . While this is a plausible deduction, the resolution is insufficient to determine what is the Co-MnO interaction that explains the catalysts activity and selectivity. The nature of the interaction, however, can be provided by DFT calculations based on a detailed knowledge of the catalyst atomic structure. Our work reveals the crucial steps in the reaction mechanism, based on the direct observation of the formation and growth of the most important intermediate CH_x, and by showing that it is formed only when Mn is present in the catalyst, mixed at the atomic scale with Co and O. Our DFT calculations, using a structural model of the CoMnO_x based on experimental data, rather than on proposed models, complete the determination of the mechanism. This is shown in energy diagrams of the rate limiting steps (**Figs. S16 to S19**).

We also read a follow-up paper from the Prof. Weckhuysen’s group: Morales, F.; Grandjean, D.; Mens, A.; de Groot, F. M. F.; Weckhuysen, B. M. X-Ray Absorption Spectroscopy of Mn/Co/TiO₂ Fischer–Tropsch Catalysts: Relationships between Preparation Method, Molecular Structure, and Catalyst Performance. *J. Phys. Chem. B* **2006**, *110* (17), 8626–8639. <https://doi.org/10.1021/jp0565958>. This paper further investigated the CoMn/TiO₂ system and states that MnO boosts C₅₊ hydrocarbon formation. The methods used, XRD and *in-situ* NEXAFS, are mostly bulk-sensitive, providing bulk rather than surface information. The authors conclude that “*These MnO/Mn_{1-x}Co_xO phases turned out to play an important role in the FT catalytic performances, leading to improvements of the C₅₊ selectivity at the expense of CH₄ production, which was always suppressed. This shift in selectivity is most likely caused by a decrease in the hydrogenation rate during the reaction, which is indicative of a more oxidized cobalt surface in which the adsorption of H₂ is suppressed and the chain growth probability is increased. Manganese in this respect decreases the amount of metallic cobalt surface, resulting in a different favorable termination path of the growing alkyl chain during FT reaction. This promotion effect is obtained as a result of an intimate interaction of the MnO species with the active cobalt particles.*” We agree with this proposed explanation, although no

experimental, or computational evidence, was provided to substantiate it. In contrast, our work using *operando* X-ray spectroscopies combined with exhaustive DFT simulations, shows that CO dissociation is promoted by the presence and reaction with H₂ (**Fig.3 and Fig.6**). We provide direct *operando* evidence for the increased production of CH_x intermediates, which occurs only in the presence of Mn (**Fig.4**). Importantly, we determined experimentally the composition and oxidation state of the topmost surface region of CoMnO_x (**Fig.2 and Fig.S6**). These structural findings provide a realistic model system for DFT calculations based on the experimentally determined structure.

(3) Dinse, A.; Aigner, M.; Ulbrich, M.; Johnson, G. R.; Bell, A. T. Effects of Mn Promotion on the Activity and Selectivity of Co/SiO₂ for Fischer–Tropsch Synthesis. *Journal of Catalysis* **2012**, 288, 104–114. <https://doi.org/10.1016/j.jcat.2012.01.008>.

(4) Johnson, G. R.; Werner, S.; Bell, A. T. An Investigation into the Effects of Mn Promotion on the Activity and Selectivity of Co/SiO₂ for Fischer–Tropsch Synthesis: Evidence for Enhanced CO Adsorption and Dissociation. *ACS Catal.* **2015**, 5 (10), 5888–5903. <https://doi.org/10.1021/acscatal.5b01578>.

These two papers are from the group of our collaborator and co-author Prof. Alexis T. Bell. The first paper focuses on the activity of CoMn/SiO₂ relative to Co/SiO₂. In the paper, the effect of the Mn:Co ratio and reactant partial pressure were investigated. The second paper includes structural characterization of CoMn/SiO₂ by TEM-EDS mapping, by Co and Mn K-edge spectra of XAS the activated catalyst, and by IR of adsorbed CO. Based on the structural characterization and the reactivity performance as function of the Mn:Co ratio, the authors proposed that “**the cleavage of the C–O bond is promoted through Lewis acid–base interactions between the Mn²⁺ cations located at the edges of MnO islands covering the Co nanoparticles and the O atom of CO adsorbates adjacent to the MnO islands.**” Our work differs from this study in the use of *operando* methods to characterize surface composition of CoMn catalysts while the FTS reaction is taking place.

(5) We have also read a related theoretical calculation paper: Chi, S.; Huang, H.; Yu, Y.; Zhang, M. Mechanism Insight into MnO for CH_x (x = 1 to 3) Hydrogenation and C1–C1 Coupling Processes on Co(0001) Surface: A DFT and KMC Study. *Applied Surface Science* **2022**, 586, 152840.

In this paper the authors use DFT and KMC methods to study C–C coupling and C–H termination steps in the FTS reaction on CoMnO_x. While we agree with the conclusions, their calculations are

based on a proposed MnO nanostructure on Co(0001). However, without experimental verification this structure may not represent the real structure of the catalyst. In our work, the model structure used for DFT calculations is based on experimental data provided by depth-profile APXPS (i.e., composition vs depth), XAS (chemical state), and by TEM imaging showing the coexistence of Co and Mn atoms within sub-nm distance, and not segregated in separate domains.

In summary, while Co and CoMn FTS catalysts have been investigated extensively, none of the cited prior work used in-situ /operando methods to obtain structural and spectroscopic insights into the role Mn^{2+} in boosting the C_{5+} hydrocarbons formation. In contrast our work shows and explains the two most important reactions steps: 1) the dissociation of CO by reaction with H_2 in operando conditions, and 2) the formation and growth of the intermediate CH_x species on $CoMnO_x$ catalysts only when Mn is present but not in its absence. Also our study provides the most extensive and detailed DFT study of the energetics and rate determining steps, using a catalyst structure based on experimental data. As an added benefit the DFT calculations were extended to explain why carbides (Co_2C) are not important ingredients of the active catalyst.

Proposed changes:

To Page 3 of the manuscript:

Line 6: "...Investigation by Weckhuysen and co-workers revealed the critical role played by $MnO/Mn_{1-x}Co_xO$ phases and by the reducible TiO_2 support in improving selectivity towards long-chain hydrocarbons.⁷⁻¹⁰”

1. The authors should explain the purpose of preparing $CoMnO_x$ thin films sample so that the readers can understand it better. According to Figure 1h and 1i, the elementary composition and distribution of NPs and thin films samples differ greatly. Additionally, the catalytic performance of thin films is absence. Whether it is appropriate to use this sample for subsequent characterization.

Reply: We thank the Reviewer for this comment. First of all, we do not think that the composition and distribution in the films and NPs differs too much. Instead we emphasize the important

similarities between the CoMnO_x crystal particles and thin films: the STEM-EDS images (Fig.1a, b) show that the 3 elements: Co, Mn and O in the NP are intimately mixed, without phase segregation after activation by H₂ (**Fig.1**). The atomic density in the NPs is similar to that in the films, as shown by the coexistence of the 3 elements: Co, Mn, O (**Fig.1d-1e and Fig.1h-1i**) in projected areas of 10 nm size, in both model catalysts.

The reviewer is correct about the lack of a catalytic performance study using the catalyst films. This is due to the impossibility to collect reaction products from films due to the insufficient amount of exposed area. In contrast, using NPs we could use large amounts of particles, with a total exposed surface area of ~6000 cm², which is 3 to 4 orders of magnitude higher than that of the catalyst films, which an exposed area of 1cm². However, in spite of this difference, **Fig.3** and **Fig.4** show that the nature of adsorbed species and their chemical evolution during FTS reaction conditions is consistent between NPs and films, as shown by the elementary steps of the FTS on CoMnO_x in thin films which are the same as those on the CoMnO_x nanoparticles.

Proposed changes:

To Page 5 of the manuscript:

Line 26: “Similar catalytic activity measurements could not be carried out on the CoMnO_x films due to the large difference in catalyst area exposed to reactants, which was ~6000 times higher for the NPs than for the films. In spite of this difference, we show in the following sections that the elementary steps of the FTS reaction are the same in the thin films and in the nanoparticles.”

2. There are two CoMnO_x NPs catalysts with a Co:Mn ratio between 1.6 and 2.8. Why the authors choose the latter to conduct subsequent characterization.

Reply: We thank the Reviewer for raising this question. As these two catalysts display quite similar product pattern of FTs reaction, we selected only one composition for our synchrotron-based XAS studies because of limited beamtime.

3. In Figure 3, the authors calculate the activation energy of H₂-assisted CO dissociation based on the assumption that the hydrogenation of carbonate species does not occur. Please provide more evidence to prove that only H₂-assisted CO dissociation happens.

Reply: We appreciate the Reviewer's insightful comment. As depicted in **Fig.3b**, after the introduction of H₂ (blue trace), two distinct O 1s XAS peaks were observed. The first peak, located around 531 eV, corresponds to lattice oxygen in CoMnO_x. The second peak, observed at approximately 534 eV, is attributed to carbonates present at the Mn²⁺ sites. This interpretation is corroborated by the C 1s APXPS results shown in **Fig.S5**, where a C 1s XPS peak at ~289.5 eV appears after the introduction of CO. It is indicative of surface carbonates. These observations suggest that the carbonates do not participate in the H₂-assisted dissociation reaction. Besides, according to our DFT calculations, the adsorption energy of CO to form carbonates, is much less likely than that of molecular adsorption on metallic Co (-0.28 eV versus -1.66 eV). So, carbonates are less stable in the basic regions of the catalyst, and they do not produce CH_x intermediates (**Fig. 4**). Therefore, they are at most spectators. We have added this discussion to the manuscript.

Proposed changes:

To Page 8 of the manuscript:

Line 22: "...showing a reduction of the barrier for H₂ induced CO dissociation on the Co-MnO surface. Besides, the H₂-assisted carbonate dissociation reaction is unlikely to occur. According to the DFT simulations, the adsorption of CO as carbonate is much less likely than molecular adsorption on the Co phase of the catalyst (-0.28 eV versus -1.66 eV). So, the carbonate is likely spectators, as any catalytic cycle involving carbonate has an activation energy of at least 1.38 eV, an energy for producing carbonate from CO. It agrees with the in-situ O 1s XAS results as after the H₂ introducing, the unreacted carbonates still remain on CoMnO_x catalyst surface."

4. In Figure 4a, the signal at 284.8 eV is assigned to CH_x species. The authors should consider the contribution of carbon deposition.

Reply: We thank the Reviewer for raising this question. We double checked the C1s XPS peak positions of several carbons located between 284.2~285.1eV (*Handbook of X-Ray Photoelectron*

Spectroscopy: A Reference Book of Standard Spectra for Identification and Interpretation of XPS Data; Moulder, J. F., Stickle, W. F., Sobol, P. E., Bomben, K. D., Chastain, J., King Jr., R. C., Physical Electronics, Incorporation, Eds.; Physical Electronics: Eden Prairie, Minn., 1995.). These C peaks are all close in energy and likely to overlap with each other. So, we agree with the reviewer that carbon deposition may contribute to the peak at ~285 eV. As we mentioned in the manuscript there is always some amount of C in that energy range due to unavoidable contamination from residual gases, even in UHV conditions. However, only the peak at ~285 eV on CoMnO_x from CH_x intermediate species grows substantially under FTS conditions, but does not occur on pure Co or pure MnO films. Also, under the same conditions a carbide peak at 283.5 eV grows on CoMnO_x and on Co metal (**Fig.4b**), while on MnO only the carbonate peak at ~289.5 eV is formed, in addition to the mentioned C contamination peak at ~285 eV. The growth of the carbide peak in CoMnO_x is only apparent, due to its overlap with the low energy side of the larger CH_x peak. On MnO_x, a carbonate peak appears in addition to the contamination peak at ~285 eV as shown in **Fig. 4b**. We are therefore confident that the increase in CH_x peak at ~285 eV on CoMnO_x (**Fig.4a**) is mostly due to the reaction CH_x intermediates.

Proposed changes

To Page 10 of the manuscript:

Line 19: “The most remarkable change is the growth of the carbon peak at ~285 eV due to CH_x species, which occurs only on CoMnO_x but not on Co or MnO. There is however a small contribution to that peak in Co and MnO due to contaminant species from background gases³⁶⁻⁴⁰ that produce species with peaks between 284~285eV. After 40 min of reaction...”

5. The authors state that H atoms being captured on O sites in MnO_x to form hydroxyls species, in agreement with the appearance of a shoulder in the XPS peak of O1s at ~531eV (Fig.S14). The authors should compare the O1s of Co sample to confirm the produced surface hydroxyl on MnO_x.

Reply: We thank the Reviewer for raising this question. Our results indicate that under the FTS conditions, a fraction of the Co is oxidized due to the CO dissociation. This oxidic Co would also be reduced by the H₂ to become metallic Co. In a previous paper (Wu, C. H.; Eren, B.; Bluhm, H.;

Salmeron, M. B. Ambient-Pressure X-Ray Photoelectron Spectroscopy Study of Cobalt Foil Model Catalyst under CO, H₂, and Their Mixtures. *ACS Catal.* **2017**, 7 (2), 1150–1157. <https://doi.org/10.1021/acscatal.6b02835>), we studied the O1s XP spectra measured under similar conditions, the O1s peak also displayed a small peak at ~531 eV, indicative of hydroxyls on the Co surface. It is really challenging to distinguish the hydroxyl on the Co surface and the MnO_x surface. Therefore, we have changed our interpretation of the O1s spectra reported in the present study.

Proposed changes

To Page 14 of the manuscript:

Line 19: “in agreement with the appearance of a shoulder in the XPS peak of O1s at ~531eV (**Fig.S14**). Hydroxyls from the Cobalt surface may also contribute to this peak.⁴⁶ Once...”

6. The authors state that CH_x adsorption on MnO is unfavorable, therefore facilitating the increase of CH_x* coverage on Co. Please give more explanation. In addition, the energy barriers of CH₂-CH₂ coupling on Co (0.38 eV) and MnO_x/Co (0.36 eV) are too close. This may not reflect the promotion effect of Mn on C-C coupling over Co catalyst.*

Reply: The increase of CH_x* coverage is confirmed by the experimental results, **Fig.4**. The theoretical explanation comes from the following three points: The conversion rate of CO to CH* increased on CoMnO_x because it has a lower energy barrier than on Co⁰ (0.98 eV versus 1.20 eV), which increases the CH* generation rate on the surface. The hydrogen atoms on Co can diffuse and bind to the basic sites in the MnO_x area thus limiting the H coverage available for the hydrogenation of CH_x (leading to termination steps) and resulting in a further increase of the CH_x coverage per Co site. Finally, CH_x* adsorption on MnO_x patches is unfavorable (**Fig.4b**), leading to a confined space for CH_x only in the metallic Co sites of the catalyst. The difference in reaction rate due to the barrier difference (0.02 eV) is indeed small as the Reviewer points out, but the confinement effect raises the coverage and number of neighboring CH_x fragments thus increasing the overall rate.

7. The authors state that H₂-assisted CO dissociation is not the RDS according to Figure 3e. However, DFT results suggested that H₂-assisted dissociation is the RDS. Please give more explanation.

Reply: The reviewer makes us realize that the text in the paper is unclear. We think that the low energy barrier derived from the fitting of the Arrhenius plot does not directly mean that the H₂-assisted dissociation is not the RDS. So, we revised this sentence to avoid misunderstandings.

Proposed changes:

To Page 8 of the manuscript:

Line 22: “An Arrhenius plot of the rate, Fig.3e, gives an activation energy of ~28 kJ/mol, a value substantially smaller than that of the typical FTS reaction (80 kJ/mol)⁸, showing a reduction of the barrier for H₂ induced CO dissociation on the Co-MnO surface.

Reviewer #2 (Remarks to the Author):

The manuscript presents a comprehensive study on the role of manganese (Mn) in CoMnO_x catalysts for the Fischer-Tropsch synthesis (FTS) reaction. The authors aim to provide insights into the molecular mechanisms underlying the enhanced selectivity for long-chain hydrocarbons on CoMnO_x surfaces by combining experimental and theoretical approaches. However, there is a noticeable gap between the theoretical predictions and the experimental observations. Addressing this gap is crucial for enhancing the robustness and reliability of the conclusions drawn from the study.

Reply: We thank the Reviewer for careful reading of our manuscript and raising critical comments. However, our experiments and DFT calculations are carefully evaluated and mutually supportive. The model used in our calculations was built following the constraints coming from the synthetic protocol and determined by the characterization: (i) a top layer that contains both Co and Mn in around a 2.5 ratio, (ii) the charge of the two metals differs, while Co keeps a metallic character Mn is present as Mn²⁺, (iii) following standard practice in modeling (taking the lowest energy structures) we employ the most compact surfaces for each of these two systems.

The representativity of the model is also confirmed as the main observations are retrieved with the simulated structures. In particular, simulations explain the experimental findings: when Mn is added to the starting Co material, (i) the coverage of CH_x increases, (ii) the activation energy of

H₂-assisted dissociation decreases, (iii) chain growth is more likely. The main take-home message is that the main building blocks of an active material require a metallic phase, (i) where the C-C bonds are formed, and (ii) chemically flexible interface that can accommodate O atoms to enhance HCO dissociation, (iii) a close contact between these two active parts, and (iv) a strict control of the H chemical potential, in this case done by the oxide as it can store the excess of protons thus increasing the probability of C-C coupling and longer hydrocarbon chains. The architecture that our model has contains all these elements, but we are sure that several other local arrangements containing these four items could provide similar results.

This work employs a relatively simple theoretical model, while the experimental characterization mainly focuses on analyzing the surface chemical composition, lacking structural information. The microstructure has a significant impact on the reaction mechanism and activity. Therefore, the reasonableness of the theoretical calculation model used in this study is worth considering, or further explanation is needed to justify the validity of the theoretical model.

Reply: We thank the reviewer for raising this question. Actually, the in-situ XRD measurements undertaken follows the bulk structural changes (**Fig.S1**), that match the TEM results in **Fig.1**. The results demonstrate that after oxidation of the crystalline CoMn nanoparticles, the bulk structure evolves from the solid solution to the spinel structure. Then after the reduction, the bulk structure changes to MnO due to segregation of Co onto the particle surface. Moreover, using depth-profiling XPS, we determined that the surface composition of activated CoMnO_x corresponds to a Co/Mn ratio of 6 in the near surface region and 2.5 in the top surface region, respectively. Another structural data is the spatial proximity of the 3 constituent atoms, Co, Mn, and O, which TEM shows coexist in small areas of nm dimension. These structural data are used to build the model used for our DFT calculations.

Considering that the theoretical calculations occupy a significant portion of this work and their prominent role in the final conclusions, I believe the manuscript does not yet meet the standards for publication. The integration of theoretical calculations and experimental characterization, as well as the justification of the theoretical model's validity—such as its thermodynamic stability and stability under reaction conditions—need to be further strengthened.

Reply: We agree that a reasonable model is crucial for correctly understanding the process, so we put a great effort in building interface models by comparing with the experimental evidence and considering the thermodynamic stability and stability under reaction conditions computationally.

The model was derived based on a several pre-tests including lattice matching, charge analysis and global optimization. For readability considerations, in the original manuscript, we mainly highlighted the oxidation and reduction of the model we choose to illustrate the stability of the structural model under reaction conditions. The whole test process is now added into the manuscript and supporting information.

The models were built following the constraints determined in the characterization: (i) a top layer that contains both Co and Mn in around a 2.5 ratio, (ii) the charge of the two metals differs, while Co keeps a metallic character Mn is present as Mn^{2+} , (iii) we employ the most compact surfaces for each of these two systems. To this end, a MnO doped Co slab was built as CoMnO_x where the topmost layer of the CoMnO_x model contains 12 Co atoms, 4 Mn atoms with O atoms at the interface.

To decide the oxygen number in the interface, we used the minima hopping algorithm to perform global optimization and sufficient oxygen atoms are provided in the interface. Around 50 ab-initio molecular dynamics simulation (AIMD) and optimization were run in the global optimization with initial thermalization temperatures of 1000 to 5000 K. In total, 26 minima were found, and the energy of the most stable one (index 24) is 0.74 eV lower than the initial structure (index 0) (**Fig.S18**). The structure shows that 5 oxygen atoms stay in the interface and the other three move to Co sites away from the interface (**Fig.S19**). In addition, considering the stability under reaction condition (**Fig.S10**), the interface with 4 oxygen is most representative, because removing one oxygen atom as H_2O is endothermic by 0.95 eV, while adding one O at the interface between Mn and Co is unfavorable by 0.15 eV.

The Bader charge analysis shows that the Mn in the model has a charge of +1.20 |e|, comparing with that of MnO (+1.47 |e| for Mn in the bulk, +1.39 |e| for Mn on the surface) and Mn_2O (+0.72 |e| for Mn in the bulk), the Mn atom in the CoMnO_x model has similar oxidation state as Mn^{2+} . The average charge of all Co atoms is +0.001 |e|, and the average charge of topmost Co atoms near MnO_x is +0.07 |e|, indicating the Co is generally metallic but slightly oxidized near the interface. Thus, considering its thermodynamic stability and stability under reaction conditions and the oxidation state, the model was used in simulation of reactions.

Fig. S18 Relative energy of the minima found using minima hopping algorithm.

Fig. S19. Configuration structures of the MnOx/Co(0001) model from minima hopping simulation. Co in pink, Mn in purple, O in red.

Proposed Changes:

To Page 11 of the manuscript:

Line 19: “And interfacial structure was determined by combining global optimization (See SI computational details) and the calculation of stability under reaction conditions (Fig.S10), as that illustrated in Fig.S10a (center) with a unit cell in the topmost layer containing 12 Co atoms, 4 Mn atoms and 4 O atoms.”

SI computational details and two figures in the end

...A K-point grid of $4 \times 6 \times 1$ was used to sample the Brillouin zone of the CoMnOx model. The number of oxygen atoms at the interface was decided via minima hopping simulation. Around 50 AIMD and optimization were run in the global optimization with initial thermalization temperatures of 1000 to 5000 K. In total, 26 minima were found, and the energy of the most stable one (index 24) is 0.74 eV lower than the initial structure (index 0), Fig.S18. The structure shows that 5 oxygen atoms stay at the interface and three move to Co sites, away from the interface, Fig.S19. Considering the stability under reaction condition (Fig.S10), the interface with 4 oxygen atoms is the most representative one and, therefore, it is used in the reaction study.

Reviewer #3 (Remarks to the Author):

In "The role of Manganese in CoMnOx catalysts in the Fischer-Tropsch reaction to enhance selectivity for long-chain hydrocarbon products" The authors measure the surface composition of working model Co/MnOx catalysts at (very) low pressures of CO and H2. The Results of the authors imply that paraffinic hydrocarbons are the main product of the reaction in their reaction conditions, which I find surprising, especially with the unusually high Manganese content (approx. 30 % rel. Co). At such high Manganese content, I would expect a high olefin selectivity and even a not negligible amount of oxygenates, at least at more technical reaction pressures (several to tens of bar).

Reply: We thank the reviewer for the comment concerning the composition of the CoMn catalyst. Fig.S2 shows catalytic performance of two representative CoMn nanoparticles for FTS. In one, the bulk mole fraction of Co is 61% and in the other is 74%. The product compositions achieved with these catalysts is similar to those reported by Dinse et al. shown below.

Figure Redacted

Dinse, A.; Aigner, M.; Ulbrich, M.; Johnson, G. R.; Bell, A. T. Effects of Mn Promotion on the Activity and Selectivity of Co/SiO₂ for Fischer–Tropsch Synthesis. *Journal of Catalysis* **2012**, 288, 104–114. <https://doi.org/10.1016/j.jcat.2012.01.008>.

The authors address this on p. 15, but only very briefly and, in my opinion, slightly to briefly. Was the composition measured by ICP-OES or AES? There is a discrepancy between the results and experimental section.

Reply: We thank for the reviewer for the comment concerning the composition of the CoMn catalyst. We used the Inductively Coupled Plasma Optical Emission spectroscopy (ICP-OES) to determine the catalyst composition. This method shows that the Co:Mn ratio is 2.76:1, corresponding to 74% Co. We have corrected this in the main draft.

Proposed Changes:

To Page 4 of the manuscript:

Line 10: “The bulk composition of the NPs was measured by Inductively Coupled Plasma Optical Emission Spectrometry (ICP-OES).”

To Page 6 of the manuscript:

Line 17: “For CoMnO_x NPs with Co:Mn ratio of 2.76:1 (from ICP-OES) ...”

The article provides insight to a possible mechanism of the Fisher-Tropsch synthesis -a topic that is still under wide debate- and contributes to the understanding of the reaction. All in all, I find it is meaningful and well written.

Reply: We thank the reviewer positive comment.

Response to the Reviewers:

Reviewer #1 (Remarks to the Author):

The authors have made very important changes and improvements in the manuscript that I believe address most of the referees' comments. I recommend acceptance in its current form.

Reply: We thank the Reviewer for their positive comment.

Reviewer #2 (Remarks to the Author):

I appreciate the authors' efforts in responding to my previous comments and the detailed explanation provided regarding the theoretical model and its validation. However, I believe the response fails to adequately address several critical points, and the manuscript still does not meet the necessary standards for publication. Below are my concerns with the authors' reply:

Reply: We sincerely thank the Reviewer for their thorough review of our manuscript and for providing valuable and insightful comments. We address Reviewer's question point by point below.

Gap Between Theory and Experiment: The authors claim that their DFT model and experimental results are mutually supportive, yet they fail to sufficiently bridge the gap between theoretical predictions and experimental observations. The theoretical models are built based on several assumptions, and the experimental data presented is insufficient to justify the accuracy and relevance of these models to real-world systems.

Reply: Experimentally, we observed that the activated CoMnO_x catalyst displays a core-shell structure, featuring a bulk retaining the spinel oxide structure (in-situ XRD results in **Fig. S1**). For the surface and near surface region, where the surface catalysis reaction occurs, we utilized two surface-sensitive methods (in-situ APXPS and the soft X-ray XAS) to determine the chemical state of the Co and Mn element, finding that Co is metallic state, and Mn is 2+ state. We used various X-ray energies to obtain the depth profile of the activated CoMnO_x, i.e., ~700 eV and ~200 eV

kinetic energy of photoelectrons bring in the Co:Mn compositional ratio from ~1.2 nm and ~0.6 nm, responding to 'near surface' and 'top surface' regions, respectively. We found that, the Co:Mn ratio is 2.5 in the top surface region and 6 in the near surface region. Our TEM measurement shows also that the spatial proximity of the 3 constituent atoms, Co, Mn, and O, is such that they coexist in small areas of nm dimension. Therefore, we built the CoMnO_x catalyst based on this structural information of the activated catalyst: (1) A top layer that contains both Co and Mn in an approximate 2.5:1 ratio, (2) the oxidation states of the two metals differ, with Co maintaining a metallic character while Mn is present as Mn^{2+} , and (3) we employ the most compact (lowest energy) surfaces for both systems.

Based on this, a MnO-doped Co slab, referred to as CoMnO_x , was constructed where the topmost layer of the CoMnO_x model consists of 12 Co atoms, 4 Mn atoms, and O atoms located at the interface. We employed the minima hopping algorithm to perform a global optimization, ensuring that sufficient oxygen atoms was present. Approximately 50 ab-initio molecular dynamics (AIMD) simulations and optimizations were conducted, with initial thermalization temperatures ranging from 1000 to 5000 K. In total, 26 minima were identified, with the energy of the most stable structure (index 24) being 0.74 eV lower than the initial structure (index 0) (**Fig. S18**). This structure reveals that 5 oxygen atoms remain at the interface, while 3 migrate to Co sites away from the interface (**Fig. S19**). In addition, considering the stability under reaction condition (**Fig. S10**), the interface with 4 oxygen is most representative, because removing one oxygen atom as H_2O is endothermic by 0.95 eV, while adding one O at the interface between Mn and Co is unfavorable by 0.15 eV. We then performed a Bader charge analysis: (1) the Mn atom in the model has a charge of +1.20 |e|, comparable to Mn in MnO (+1.47 |e| for bulk Mn, +1.39 |e| for surface Mn) and Mn_2O (+0.72 |e| for bulk Mn), suggesting that Mn in the CoMnO_x model has a similar oxidation state as Mn^{2+} ; (2) The average charge of all Co atoms is +0.001 |e|, while the average charge of the topmost Co atoms near MnO_x is +0.07 |e|, indicating that Co remains largely metallic, with slight oxidation near the interface. In summary, we believe that based on experimental structural information, and further examined by AIMD and Bader charge analysis, the structural model we used, is suitable to explain the reaction mechanism.

The authors state that they employed the Co(0001) surface in their calculations based on "(iii) following standard practice in modeling (taking the lowest energy structures) we employ the most compact surfaces for each of these two systems." While Co(0001) may indeed be the most stable surface, it is widely recognized that Co(0001) is unlikely and/or not necessarily true to serve as the active site for Fischer-Tropsch synthesis. Therefore, comparing reaction activity and selectivity results to Co(0001) remains a great uncertainty. The suitability of constructing the CoMnO_x model based on Co(0001) needs to be reconsidered more carefully, not only relying on the authors' own experimental characterization but also incorporating the broader understanding of active sites in cobalt catalysts for FTS from past research. This broader perspective is essential to ensure that the chosen model is representative of actual catalytic behavior in FTS.

Reply: We appreciate the Reviewer for raising this critical comment regarding the Co active phase. We agree that the activated facet of Co is still a matter of debate. In our own experimental results, we are unable to determine definitively the exposed facet of the activated CoMnO_x catalyst, as the nanoparticles remain in a round shape. However, according to the “Wulff construction”, the equilibrium shape of nanoparticles tends to have a higher area ratio of the facet with the lowest surface energy. For Co, the (0001) facet has the lowest surface energy, suggesting that this facet is more accessible compared to others. This explains why most theoretical and experimental studies on Co-based Fischer-Tropsch synthesis reactions focus on the hcp(0001) facet or fcc(111), these two surfaces exhibit identical hexagonal geometry and comparable lattice parameters. The only structural variation between them occurs at the third Co layer. As a result, the adsorption energetics on these densely packed surfaces are expected to be nearly identical. (ACS Catal. 2012, 2, 6, 1097–1107) ¹ Some experimental studies using Co model catalysts with other terminations, such as in references (11-20) and (10-12), have shown that molecular CO can induce the migration of Co atoms under ultra-high vacuum conditions and room temperature, suggesting surface reconstruction.²⁻⁴ It is expected that under real conditions (high temperature and ambient pressure), this reconstruction would become even more pronounced as the chemical potential of CO increases significantly. In contrast, the Co(0001) surface remains more stable under reaction conditions, allowing high-temperature and high-pressure in-situ STM measurements^{5,6}. In the present study we focus on understanding the effects of Mn on the reaction, rather than investigating the performance of the Co surface or comparing it to other metals or orientations. To achieve this, the

Co surface is kept consistent across different models, with the pure Co surface serving as a control sample, although the Co orientation might change as seen in other FTS studies the key structural features regarding the interface are local in nature and these are reasonably captured in our model. Therefore, using the Co(0001) facet in our model is rational in the present work. Although we agree that further exploration could improve determining the real active surface of Co under reaction conditions, this is beyond the scope of this paper.

Proposed changes:

To Computational details of the SI:

“A 4 layer slab of p(4×4) supercell, with the bottom 2 layers fixed, is used to model the Co(0001) with k-point grid of 5×5×1 to sample the Brillouin zone. A 4 layer slab of p(2×2) supercell with bottom 2 layers fixed are used to model the MnO(100) with k-point grid of 3×3×1. We acknowledge that other Co facets may also contribute to the Fischer-Tropsch synthesis reaction. However, to specifically investigate the critical role of Mn, we simplified the structural complexity by selecting the Co(0001) facet, which has the lowest surface energy, as the starting structural model.”

2、 Structural Considerations: While the authors considered the structural information from XRD and TEM, there is still a lack of sufficient structural characterization, particularly regarding the microstructure of the catalyst and how it influences the reaction mechanism. The reply provides some justification for the theoretical model, but it does not fully resolve the concerns about the structural complexity of the catalyst, which may play a critical role in its catalytic behavior. The authors claim that the "top surface regions" are approximately 0.6 nm thick, which actually corresponds to 2-3 atomic layers. However, characterization of the "top surface regions" alone is insufficient justification for simplifying the model to this extent. Multiple potential interfacial configurations should have been considered, rather than simply replacing two rows of Co atoms with MnO on the (0001) surface. Furthermore, the choice of the (0001) surface itself is questionable, as it is not representative of the active sites typically associated with Fischer-Tropsch synthesis. The current model appears somewhat hasty and lacks the thoroughness needed to accurately capture the complexity of the catalyst's structure and behavior. A more detailed

exploration of possible interface configurations would be necessary to ensure a robust and representative model.

Reply: We appreciate the Reviewer for raising this critical comment on the structural model. We acknowledge, it is very challenging to really determine the exposed facet of the activated CoMnO_x catalyst in the round shape. As discussed in the above reply, the “Wulff construction” theorem should be applicable to the nano-catalysts investigated here, which are in equilibrium with the surrounding reaction gases and temperature. Therefore, we believe that the (0001) facet is structurally reasonable as other less stable facets Co would undergo significant surface reconstructions.

Moreover, the process of selecting the model is more complex than merely replacing two rows of Co with MnO. As mentioned earlier, this model was systematically validated through global optimization and thermodynamic calculations, now added to the SI. Additionally, several possible interface configurations were constructed prior to the global optimization, as described below. A previous theoretical study used a $\text{Co}(0001)$ surface and a supported MnO ribbon to simulate the CoMnO system (Applied Surface Science 567 (2021) 150854). This model was also considered in our model selection process. However, further investigation revealed that the ribbons could aggregate (**Fig. R1**) to form larger MnO ribbons with a formation energy of -1.77 eV, potentially covering all active Co sites and thus breaking the experimentally observed Co-to-Mn ratio of 2.5:1. Therefore, this model was not considered for further simulations.

Fig R1. The aggregation of support ribbon model. (a) before aggregation, (b) after aggregation.

Thus, models embedding MnO into the topmost Co surface were considered. Two of the models were constructed to evaluate the compatibility between MnO and Co, **Fig R2**, by removing either two or three rows of Co. After relaxation, it was clear that the structure of MnO and the subsurface Co was not maintained in the later model, indicating the instability of the model. Subsequently, the model with two rows of Co replaced by MnO was employed and further validated. It is worth noting that some models that emerged from the global optimization process, such as those numbered 14-17 (**Fig. S19**), deviated significantly from the simple model. However, these structures proved to be significantly less stable than the one we ultimately used.

Fig. R2 The model with different Co atoms removal. (a) removing 2 rows, (b) removing 3 rows. The top view is at the top and the side view is at the bottom.

Fig.S19 The structure of the $\text{MnO}_x/\text{Co}(0001)$ model from minima hopping simulation. Co in pink, Mn in purple, O in red.

While we acknowledge that a large number of microstructures could exist, sampling all of them exceeds current state-of-the-art approaches. However, by focusing on the most likely configurations, we developed a model that aligns well with experimental structural results, providing insight into the key scientific question: “how the Mn can boost the C_{5+} hydrocarbon products”. Moreover, the structural model’s stability was further assessed by AIMD method, and

the results positively demonstrate a suitable range of stability. Therefore, we are confident about the structural model that we used.

In conclusion, while the authors have made some effort to clarify their approach, significant issues remain with the integration of experimental and theoretical work. I recommend that the authors strengthen the connection between their theoretical predictions and experimental results, provide more experimental validation for the chosen model, and address the concerns regarding the structural complexity of the catalyst. Without these improvements, the manuscript is not ready for publication.

Reply: We hope the above explanations provide clarity and satisfy the Reviewer's concerns.

Reviewer #3 (Remarks to the Author):

The authors seem to have reacted to the critiques and have implemented according changes in the manuscript. In my opinion the manuscript is publishable.

Reply: We thank the Reviewer for their positive comment.

Response to the Reviewers:

Reviewer #2 (Remarks to the Author):

The authors have provided sufficient explanations and clarifications regarding the issues I was concerned about. I find their responses satisfactory. Therefore, I recommend accepting the manuscript.

Reply: We sincerely thank the Reviewer #2 for the thorough review of our manuscript and for providing valuable and insightful comments. We appreciate your time and feedback on our manuscript